# Biosynthetic Silver Nanoparticles Inhibit the Malignant Behavior of Gastric Cancer Cells and Enhance the Therapeutic Effect of 5-Fluorouracil by Promoting Intracellular ROS Generation and Apoptosis

**DOI:** 10.3390/pharmaceutics14102109

**Published:** 2022-10-02

**Authors:** Jingwen Yuan, Shahid Ullah Khan, Jiajun Luo, Yue Jiang, Yu Yang, Junfeng Yan, Qiang Tong

**Affiliations:** 1Department of Gastrointestinal Surgery I Section, Renmin Hospital of Wuhan University, Wuhan 430060, China; 2Department of Biochemistry, Women Medical and Dental College, Khyber Medical University, Abbottabad 22080, Pakistan

**Keywords:** gastric cancer, biosynthetic, chemotherapeutic drug, therapy, proliferation, migration, invasion, mechanism

## Abstract

(1) Background: Gastric cancer (GC) is the fourth leading cause of cancer death worldwide. Silver nanoparticles (Ag-NPs) have been increasingly used in the diagnosis and treatment of cancer due to their physicochemical properties. This study investigated the role of a kind of biosynthetic silver nanoparticle (b-Ag) in the development of GC, the enhancement of 5-fluorouracil (5F), and its mechanism. (2) Methods: X-ray, transmission electron microscopy (TEM), and UV absorbance were used to detect the characterizations of AgNPs. CCK8, Colony formation and a Transwell assay were performed to confirm the malignant behaviors of GC. DCFH-DA and DHE were used to detect intracellular reactive oxygen species (ROS). Quantitative reverse transcription polymerase chain reaction (qRT-PCR) was used to detect the mRNA expression of apoptosis-related genes. (3) Results: Compared with the chemosynthetic silver nanoparticles (c-Ag), b-Ag had a stronger cytokilling effect, and it had a better inhibition on the malignant phenotype of GC when combined with 5F. The b-Ag increased the expression of Bax and P53 while decreasing the expression of Bcl2. It also promoted the generation of intracellular ROS. (4) Conclusions: By promoting cell apoptosis and increasing intracellular ROS, b-Ag inhibited the development of GC and enhanced the inhibition of 5F on GC.

## 1. Introduction

Gastric cancer (GC), one of the most common cancers, has become the fifth most diagnosed cancer and the fourth leading cause of cancer death [1].

Nanoparticles, as emerging biotechnology, play a key role in fighting against breast cancer [2], ovarian cancer [3], prostate [4], and other cancers due to their unique physical and chemical properties, and have also attracted more and more attention in anticancer therapy due to their good biocompatibility [5], as well as their ability to carry drug particles [6], and target specific cells [7]. Metal nanoparticles (especially silver nanoparticles), as one of the most common nanoparticles, have been widely used in the treatment of drug-resistant bacteria [8], mastitis [9], the diagnosis and treatment of various cancers [10,11,12], and disease tracer [13], but there are few studies in GC.

5-Fluorouracil (5F) is a common chemotherapeutic agent for gastrointestinal cancer. It can inhibit thymidylate synthase and suppress DNA synthesis. It also has a slight inhibitory effect on RNA synthesis, and ultimately affects the metabolism of cells, inhibiting the proliferation of cells [14]. Still, it has side effects that cannot be ignored: chemotherapy-induced diarrhea [15], intestinal mucositis [16], hand–foot syndrome [17], and coronary Vasospasm [18], the most important of which is myelosuppression [19]. Therefore, it is essential to find a drug to assist 5F, so that 5F can have a high-killing effect on gastric cancer cells at a low concentration.

In this paper, we described a process in which a kind of biosynthetic silver nanoparticles synthesized from *Atropa acuminata Royle Ex Lindl* extract as a reducing agent inhibited the GC malignant behavior and enhanced the killing effect of 5F on GC by promoting apoptosis and producing excessive intracellular reactive oxygen species (ROS).

## 2. Materials and Methods

### 2.1. Cell Culture

Human gastric cancer cells MGC-803 and HCG-27 were purchased from the American Type Culture Collection (ATCC, Manassas, VA, USA). Cells were cultured in DMEM (Gibco, Waltham, MA, USA, Dulbecco’s Modified Eagle Medium-High Glucose) supplemented with 10% fetal bovine serum (Gibco, Waltham, MA, USA, FBS) and 1% dual antibiotics (penicillin + streptomycin) at a temperature of 37 °C with 5% CO_2_. The cells were cultured to the logarithmic growth phase and digested with 0.25% trypsin. After 3 min, the cells were observed under the microscope to turn around, indicating that digestion could be stopped. The DMEM with 10% FBS was added to stop digestion, and we blew and mixed the cell suspension to make it singular. The number of cells were counted, then we proceeded to the next step.

### 2.2. Materials

The biosynthetic silver nanoparticles and chemically synthesized silver nanoparticles were provided by Dr. Shahid Ullah Khan of the Department of Biochemistry, Women’s Medical and Dental College Abbottabad, KPK-Pakistan. The biosynthetic silver nanoparticles were made using a special kind of biological material as reductant, which was extracted from *A. acuminata* leaf (*A. acuminata* leaves were extracted by methanol after they were cleaned by Millipore water and finally dried into a powder. Every 1 mL 20% dimethyl sulfoxide aqueous solution contains 10 mg powder.). This biological material was hereinafter referred to as AX. AX and AgNO_3_ (10^−2^ M) were mixed in a ratio of 500:150 (μL) and dissolved in 4.35 mL deionized water to prepare biosynthetic silver nanoparticles (Abbreviate to b-Ag). The chemically synthesized silver nanoparticles were made using chemosynthetic material as reductant, in a mixture of 2.8 mL NaBH_4_ (0.05 mg/mL) and 150 μL AgNO_3_ (10^−2^ M). We referred to this as CX. Additionally, the precipitates were collected after centrifugation at 14,000 rpm for 1 h at 10 °C. We dissolved the precipitate with deionized water, and then obtained chemically synthesized silver nanoparticles (Abbreviate to c-Ag). X-ray diffraction (XRD) measurement was performed using the Rigaku D-Max 2200 X-ray diffraction system with a Cu Kα radiation at 35 kV and 35 mA. Transmission electron microscopy was performed on the FEI Tecnai G2 F300 instrument. The UV-absorbance spectra of the AgNPs were recorded on a UV-vis-NIR spectrophotometer (Shimadzu UV-3600). The 5F was provided by the Department of Gastrointestinal Surgery I Section, Renmin Hospital of Wuhan University. The drug was a white powder; it must be kept in cool, dry places away from direct sunlight. We dissolved 34 mg 5F powder (molecular weight 342.86) per 1 mL PBS to prepare the 100 mM 5F solution, then diluted it with PBS to the desired concentration according to experimental requirements. The 5F was filtered with a bacterial filter before use.

### 2.3. Cell Counting Kit-8 (CCK8)

After cell counting, the cells were diluted to 2 × 10^4^/mL with 10% FBS DMEM. After blowing and mixing, the cell suspension was inoculated in a 96-well plate at the volume of 100 μL/ well. The cells were divided into groups according to the experimental needs, with 5 repeat wells in each group. We added 10 μL CCK8 solution (Servicebio, Wuhan, China) to each well, shook well using the cross method and put it into an incubator at 37 °C for culture. We took out out the 96-well plate 3 h later, shook it with the microplate tester at medium speed for 1 min, and then measured the absorbance of each well at OD450 nm. Cell survival rate was calculated: Cell survival rate % = (AS − AB)/(AC − AB) × 100%, As = absorbance of an experimental well (including cells, medium, CCK8, and the compound to be tested), Ab = absorbance of a blank well (including medium, CCK8 with no cells), Ac = absorbance of control well (including cells, medium, and CCK8).

### 2.4. Colony Formation Assay

After cell counting, the cells were diluted to 250/mL with 10% FBS DMEM. After blowing and mixing, the cell suspension was inoculated in a 6-well plate at the volume of 2 mL/well. Each cell was inoculated with a 6-well plate. The culture was stopped when the number of monoclonal colonies was greater than 50, which was observed two weeks later. We discarded the medium, washed the plate three times with PBS, added 4% paraformaldehyde 2 mL/hole and set for 20 min, then abandoned the fixed liquid, adding 0.1% crystal violet dye solution 2 mL/hole dyeing and leaving it to set for 20 min after washing it three times with PBS. We removed the dye solution, cleaned it with clear water three times (4 mL/hole, 5 min/times), and finally put a 6-well plate on the whiteboard to get the result.

### 2.5. Transwell

Migration: Cells were counted with FBS-free DMEM and diluted to 10^6^/mL. After blowing and mixing, the cell suspension was inoculated into the upper chamber of the 24-well Transwell plate (Costar, Washington, WA, USA, #3422, polycarbonate membrane micropore diameter 8 μm) at the volume of 100 μL/ well. We added 10% FBS DMEM 800 μL/ well into the lower chamber, then carefully put upper chambers into the corresponding lower chambers, making sure not to produce any bubbles. After 24 h of cultivation in a 37 °C incubator, the 24-well Transwell plate was taken out, and the medium in the upper and lower chambers was carefully discarded and cleaned with PBS three times. Then, 4% paraformaldehyde was added to the lower chamber, and the corresponding upper chamber was immersed for 20 min. Then, we cleaned the upper chambers with PBS three times after paraformaldehyde was discarded. Finally, we stained the upper chamber bottom membrane with 0.1% crystal violet dye for 20 min, then removed the dye, and cleaned the upper chambers three times with clear water (5 min/time). We carefully erased the cells inside the upper chamber basement membranes using cotton swabs. After drying the upper chambers, the cells stained purple on the upper chamber bottom membranes were observed under an inverted microscope at 200× magnification, then photographed and counted. Invasion: BD Matrigel (Corning, Corning, NY, USA, #356234) was diluted by FBS-free DMEM at a ratio of 1:10, and spread evenly on the upper chamber basement membranes with the amount of 100 μL/ well. After laying, we put the upper chambers into the corresponding lower chambers. The 24-well Transwell plate was placed in an incubator at 37 °C to wait for Matrigel to solidify and taken out after 24 h. Subsequent procedures were the same as Migration procedures.

### 2.6. DCFH-DA

After cell counting, the cells were diluted to 0.6 × 10^6^/mL with 10% FBS DMEM. After blowing and mixing, the cell suspension was inoculated in a 6-well plate with a volume of 2 mL/well. Each kind of cell was inoculated in a 6-well plate, and the six wells were treated differently. After 3 days, the 6-well plates were taken out and observed under a microscope. The culture was stopped when the cells in the blank group showed logarithmic growth. DCFH-DA (Beotime, Shanghai, China, #S0033S) was diluted into a working solution with FBS-free DMEM at 1:1000, and 500 μL of the working solution was added to each well. Then, the 6-well plates were placed in an incubator at 37 °C for 20 min. After the 6-well plates were removed, the working solution was discarded, and FBS-Free DMEM was used to clean the plate three times (five min/time). Finally, the 6-well plates were observed and photographed under a fluorescence microscope under blue light excitation at 200× magnification.

### 2.7. Dihydroethidium (DHE)

After cell counting, the cells were diluted to 0.6 × 10^6^/mL with 10% FBS DMEM. After blowing and mixing, the cell suspension was inoculated in a six-well plate with a volume of 2 mL/well. Each kind of cell was inoculated in a 6-well plate, and the six wells were treated differently. After 3 days, the 6-well plates were taken out and observed under a microscope. When the cells in the blank group showed logarithmic growth, the culture was stopped. DHE (Yeason, Shanghai, China, Dihydroethidium) was diluted with PBS at 1:1000 into a working solution; 500 μL of the working solution was added to each well. The 6-well plates were incubated at 37 °C for 20 min. After the 6-well plates were removed, the working solution was discarded, and the plates were cleaned with PBS three times (five min/time). Finally, the 6-well plates were observed and photographed under a fluorescence microscope under green light excitation at a magnification of 200×.

### 2.8. Quantitative Real-Time PCR (qRT-PCR)

Total RNA of cells was extracted by Trizol (Sigma-Aldrich, St. Louis, MI, USA) according to the instructions. Then, RNA was reverse-transcribed into cDNA using 2 μL total RNA as a template using a reverse transcription kit (Vazyme, Nanjing, China, #R212-01) according to the instructions. Dilute the cDNA with ddH2O to 100 μL, then real-time PCR was performed using ChamQ SYBR qPCR Master Mix (Vazyme, Nanjing, China, #q311-02) with 2 μL cDNA as template. Reaction procedure: 95 °C 30 s, then 40 cycles of 95 °C 10 s and 60 °C 35 s, then 25 °C 1 min. The primers were as follows: P53 FP 5′-TAACAGTTCCTGCATGGGCGGC-3′, RP 5′-AGGACAGGCACAAACACGCACC-3′; Bcl2 FP 5′-CGACTTCGCCGAGATGTCCAGCCAG-3′, RP 5′-ACTTGTGGCCCAGATAGGCACCCAG-3′; Bax FP 5′-AGGGTTTCATCCAGGATCGAGCAG-3′, RP 5′-ATCTTCTTCCAGATGGTGAGCGAG-3′; Gapdh P 5′-GAAGGTGAAGGTCGGAGTC-3′, RP 5′-GAAGATGGTGATGGGATTTC-3′.

### 2.9. Statistic Analysis

The unpaired *T* test was used for comparison between groups, and *p* < 0.05 was considered statistically significant. IBM SPSS Statistics 23.0 and Graphpad Prism 8.0 were used for data analysis and mapping. Image J was used to calculate cells and measure fluorescence intensity.

## 3. Results

### 3.1. Physicochemical Characterization of AgNPs

The XRD patterns of the AgNPs nanoparticles were analyzed to identify their phase composition. From the obtained XRD patterns (Figure 1a), it was evident that the detected diffraction peaks were consistent with the standard face centered cubic AgNPs (JCPDS# No. 04-0783 for AgNPs and No. 42-0874 for silver oxide). The nanostructure and particle size of the AgNPs were evaluated via the TEM images. The TEM images, shown in Figure 1a–c, displayed that the resultant compounds were composed of uniform nanoparticles and the particle size was distributed in the range of 10–75 nm. The particle size had been measured by calculating about 70 nanoparticles. The particles size of AgNPs varied from smallest (6.5 nm) to largest of about 72 nm, with average size of 38.9 nm (Figure 1d). Additionally, Figure 1e displayed the UV visible spectra of AgNPs. The appearance of the absorption peak around 408–430 nm in UV visible spectra suggested the formation of spherical silver nanoparticles.

### 3.2. Cytotoxic Effect of AgNPs on Gastric Cancer Cells

According to Mukherjee, S. et al.’s article [20], we diluted b-Ag and c-Ag in deionized water to concentrations of 64.5, 193.5, 322.3, 644.6 and 1933.7 (PPm) for the CCK8 experiment to compare the effects of biological nano-silver and chemical nano-silver. Additionally, the concentration of b-Ag was shown in Table 1, and the concentration of c-Ag was shown in Table 2. The results showed that the cell survival rates of HCG-27 (Figure 2a) and MGC-803 (Figure 2c) decreased with the increase in b-Ag concentration, and the same trend was found in the experiment of c-Ag (Figure 2b,d). Additionally, the survival rate of the b-Ag group was lower than that of c-Ag group at the same concentration (Figure 2a–d).

Therefore, we can say that b-Ag has a better inhibitory effect on gastric cancer cell viability than c-Ag, and this inhibition is concentration dependent.

### 3.3. Enhancement of 5F Cytotoxicity in Gastric Cancer Cells by AgNPs

We used PBS to configure 5F at concentrations of 64.5, 193.5, 322.3, 644.6, 1933.7, 3222.9, and 6445.8 (PPm) for the CCK8 experiment. The concentration of 5F was shown in Table 3. The results showed that the survival rate of HCG-27 (Figure 3a) and MGC-803 (Figure 3b) was decreased with the increase in 5F concentration, and we found that when the 5F concentration reached 644.6 (PPm), the survival rate of both cells was decreased significantly and reached below 50%. Therefore, we chose 644.6 (PPm) as the concentration of 5F for the following experiment.

Figure 2a,c showed that when the concentration of b-Ag reached 322.3 (PPm), the survival rate of both kinds of cells declined significantly. Therefore, we chose the concentration of 322.3 (PPm) as the concentration of b-Ag for subsequent experiments, and the same concentration 322.3 (PPm) was selected in c-Ag for subsequent experiments. The full names and drug concentrations were shown in Table 4.

CCK8 results showed that in HCG-27 (Figure 3c) and MGC-803 (Figure 3d), when b-Ag and 5F were added to cells, the cell survival rate was lower than that of 5F group (*p* < 0.05). The cell survival rate of 5F + b-Ag group was lower than that of 5F + c-Ag group and b-Ag group.

These results suggest that b-Ag can significantly enhance the inhibitory effect of 5F on gastric cancer cell viability, and the enhanced effect is better than that of c-Ag.

### 3.4. Malignant Phenotype of Gastric Cancer Cells under AgNPs and 5F

We divided the cells into six groups, which were control, 5F, b-Ag, 5F + b-Ag, c-Ag, and 5F + c-Ag. The concentrations of groups were shown in Table 4. Corresponding drug treatments were added to the cells for co-culture. Colony formation assay and Transwell assay were used to investigate the specific role of b-Ag in gastric cancer development. The results of the Colony formation assay showed that the cell growth of c-Ag group in HCG-27 (Figure 4a) and MGC-803 (Figure 4b) was similar to that of the control group, the cell growth of b-Ag group was lower than that of c-Ag group, and when 5F was added, the cell growth decreased significantly. The effect of 5F + c-Ag was similar to that of 5F alone, while the cell growth of 5F + b-Ag was significantly reduced compared with that of 5F alone. These results indicate that b-Ag has a better inhibitory effect on gastric cancer cell proliferation than c-Ag, and b-Ag can significantly enhance the inhibitory effect of 5F on gastric cancer cell proliferation.

Subsequently, in the Transwell experiment, the migration ability of HCG-27 (Figure 4c) did not change significantly under c-Ag treatment but was close to that of b-Ag treatment. When 5F was added, the migration ability decreased significantly, and the effect was close to the result of 5F + c-Ag group. However, when the cells were treated with 5F and b-Ag, the migration ability of the cells was greatly reduced, which was reflected by the relative cell number (Figure 4d). The same assay was performed in MGC-803, and a similar trend in HCG-27 was confirmed (Figure 4g,h). In the invasion results, the number of cells successfully crossing the basement membrane of the upper chambers decreased significantly compared with the migration experiment because there was Matrigel on the basement membrane. However, the invasion trend consistent with migration can still be found in HCG-27 (Figure 3e,f) and MGC-803 (Figure 4i,j).

Therefore, it can be concluded that b-Ag has a better inhibitory effect on the proliferation, migration, and invasion of gastric cancer cells than c-Ag at the same concentration. Meanwhile, b-Ag can significantly enhance the inhibitory effect of 5F on the malignant behavior of gastric cancer cells, and the enhanced effect is much better than that of c-Ag.

### 3.5. Gastric Cancer Cell Apoptosis under AgNPs and 5F

qRT-PCR was used to detect the mRNA expression levels of Bax, Bcl2, and p53 in cells of each group. The results showed that the gene expression levels in all experimental groups were significantly altered in both HCG-27 and MGC-803 cells. The 5F + b-Ag group had the largest change, and the change in the b-Ag group was significantly greater than that in the c-Ag group. Specifically, the expression of Bax (Figure 5a) and p53 (Figure 5e) was increased, and the expression of Bcl2 was decreased (Figure 5c). The same trend was observed in MGC-803 cells (Figure 5b,d,f).

High levels of Bax and p53 suggest apoptosis, and Bcl_2_ is an antagonistic gene of Bax. Therefore, it is concluded that b-Ag promotes apoptosis and assists 5F in promoting apoptosis.

### 3.6. Intracellular ROS of Gastric Cancer Cells under AgNPs and 5F

DCFH-DA and DHE probes were used for fluorescence staining to investigate the changes in intracellular ROS in each group. Green fluorescence and red fluorescence intensity represent the content of intracellular hydrogen peroxide (H_2_O_2_) and superoxide anion (O_2_·−), respectively. The results showed that, in HCG-27 (Figure 6a,e) and MGC-803 (Figure 6c,g), the mean intracellular fluorescence intensity in the b-Ag group was higher than that in the control group and the c-Ag group, and the fluorescence intensity in the 5F + b-Ag group was higher than that in both the 5F and 5F + c-Ag groups. The quantification of their results is shown in Figure 6b,d,f,h. These results indicate that b-Ag can better promote intracellular ROS production in gastric cancer cells and induce a large amount of intracellular ROS production when combined with 5F.

## 4. Discussion

As a kind of cancer with high morbidity and mortality worldwide, the diagnosis and treatment of gastric cancer have always been the focus of researchers around the world. Common treatments include surgery, chemotherapy, radiation therapy, immunotherapy, etc. As a commonly used chemotherapeutic agent for gastrointestinal tumors, 5F has a good killing effect on gastric cancer; however, it also has inevitable side effects, such as vomiting, loss of appetite, bone marrow suppression, etc. [21], because 5F kills both tumor cells and normal cells. Severe side effects sometimes make patients unable to tolerate chemotherapy. Reducing the concentration and dose of chemotherapy drugs can alleviate the side effects to some extent, but the therapeutic effect cannot reach the expectation simultaneously. Therefore, it is necessary to find an adjunctive drug for 5F in clinical practice which can reduce the concentration and dose of 5F while ensuring the death of cancer cells. At this time, we are concerned about metal nanoparticles, which are metal particles smaller than 100 nm in size. Chemical and biological methods can synthesize them. Chemically synthesized silver nanoparticles are easy to obtain, and biosynthetic silver nanoparticles are green and environmentally friendly. Due to their excellent biocompatibility, drug delivery, cell targeting, and other properties, they have been widely studied in the tracing, diagnosis, and treatment of cancer, as well as the delivery of chemotherapy drugs [22,23,24,25]. However, the study of silver nanoparticles in gastric cancer has rarely been reported. Therefore, the effects of two kinds of silver nanoparticles on gastric cancer cells and the auxiliary of silver nanoparticles on 5F were studied in this paper.

Through the CCK8 experiment, we confirmed that b-Ag has a better killing effect on gastric cancer cells than c-Ag at the same concentration. After finding a 50% inhibiting concentration of 5F, the two AgNPs were combined with 5F, respectively, and a satisfactory killing effect was observed when b-Ag was combined with 5F. Subsequently, the colony formation experiment and Transwell experiment were used to confirm that b-Ag combined with 5F could effectively inhibit the proliferation, migration, and invasion of gastric cancer. Since 5F can kill cancer cells by promoting apoptosis [14], we detected the expression of Bax, Bcl_2_, and p53 in cells under different treatments. The results showed that b-Ag increased the expression of Bax and p53 while decreasing the expression of Bcl_2_, and the changes in expression were more significant when b-Ag was combined with 5F. Bax and Bcl_2_ antagonize each other, and p53 can regulate the Ratio of Bax/Bcl_2_. When the ratio of Bax/Bcl_2_ increased, the caspase3 pathway was activated, and p53 was also highly expressed at that time. More cells entered the G0 phase, and apoptosis increased [26,27]. Therefore, we speculated that the killing effect of b-Ag on gastric cancer and the assisting effect of b-Ag on 5F might be achieved through promoting cell apoptosis. Oxidative stress, a pathway of apoptosis of various cancer cells [28,29], has also been verified in gastric cancer cells. We found that the intracellular ROS amount was increased in cells treated with b-Ag and 5F at a lower concentration.

Therefore, we hypothesized that b-Ag had a better killing effect and enhanced the cytotoxicity of 5F on gastric cancer more than c-Ag, by increasing intracellular ROS production, promoting apoptosis, and increasing the ratio of cells in the G0/G1 phase. In the future, silver nanoparticles, including other metal nanoparticles, such as gold and copper nanoparticles, still have many other pathways and functions, such as red blood cell membrane-camouflaged nanoparticles [30], which we need to explore.

## 5. Conclusions

Biosynthetic silver nanoparticles inhibit the malignant behavior of gastric cancer cells and enhance 5F in suppressing the growth, migration, and invasion of gastric cancer cells by promoting apoptosis and increasing intracellular ROS production.

## Figures and Tables

**Figure 1 pharmaceutics-14-02109-f001:**
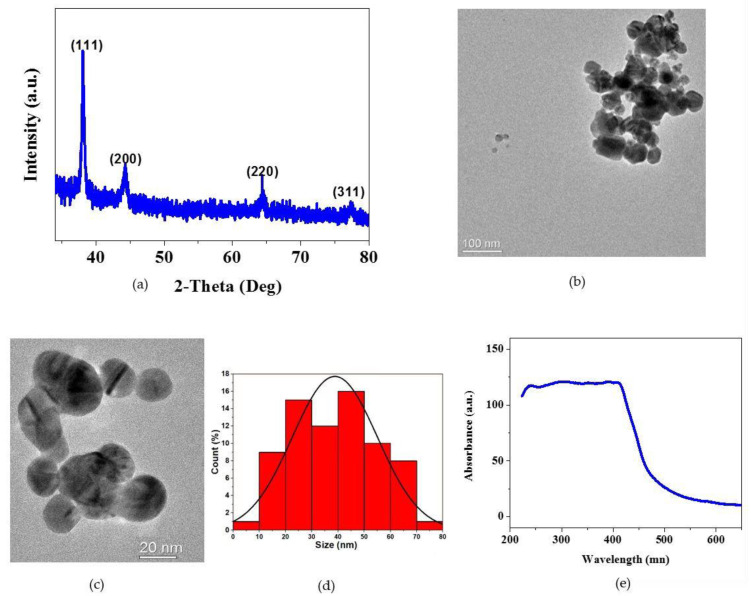
Physicochemical characterization of AgNPs. (**a**) X-ray diffraction pattern (XRD) of AgNPs demonstrates the crystalline nature of silver nanoparticles; (**b**–**d**) TEM indicates the morphology of AgNPs and size of the particle; (**e**) UV absorbance of AgNPs.

**Figure 2 pharmaceutics-14-02109-f002:**
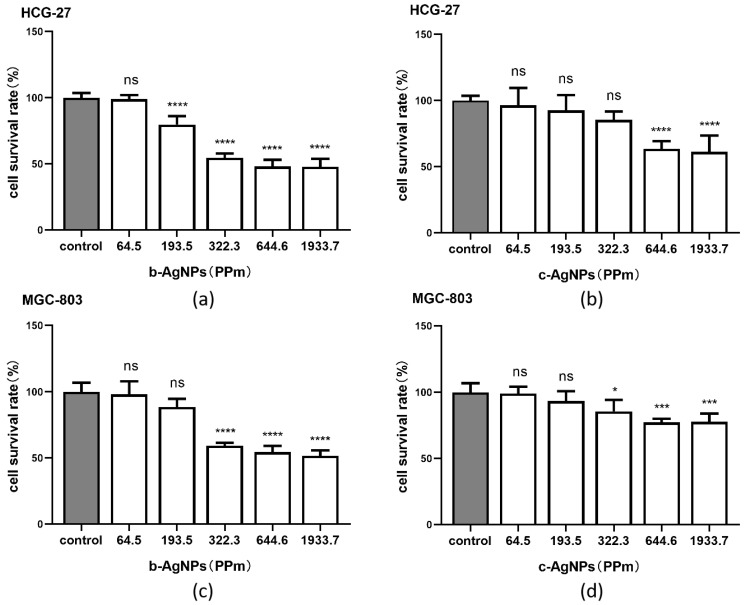
Cell survival rate under different concentrations of silver nanoparticles. (**a**) The cell survival rate of HCG-27 under different concentrations of b-Ag; (**b**) cell survival rate of HCG-27 under different concentrations of c-Ag; (**c**) cell survival rate of MGC-803 under different concentrations of b-Ag; (**d**) cell survival rate of MGC-803 under different concentrations of c-Ag. * *p* < 0.05, *** *p* < 0.001, **** *p* < 0.0001.

**Figure 3 pharmaceutics-14-02109-f003:**
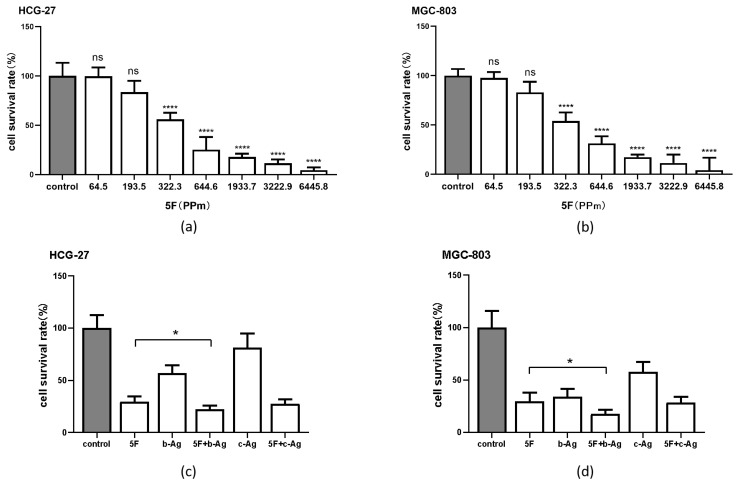
The cell survival rate of 5F alone and AgNPs combined with 5F. (**a**) The cell survival rate of HCG-27 at different concentrations of 5F; (**b**) cell survival rate of MGC-803 at different concentrations of 5F; (**c**) cell survival rate of HCG-27 under 5F + AgNPs; (**d**) cell survival rate of MGC-803 under 5F + AgNPs. * *p* < 0.05, **** *p* < 0.0001.

**Figure 4 pharmaceutics-14-02109-f004:**
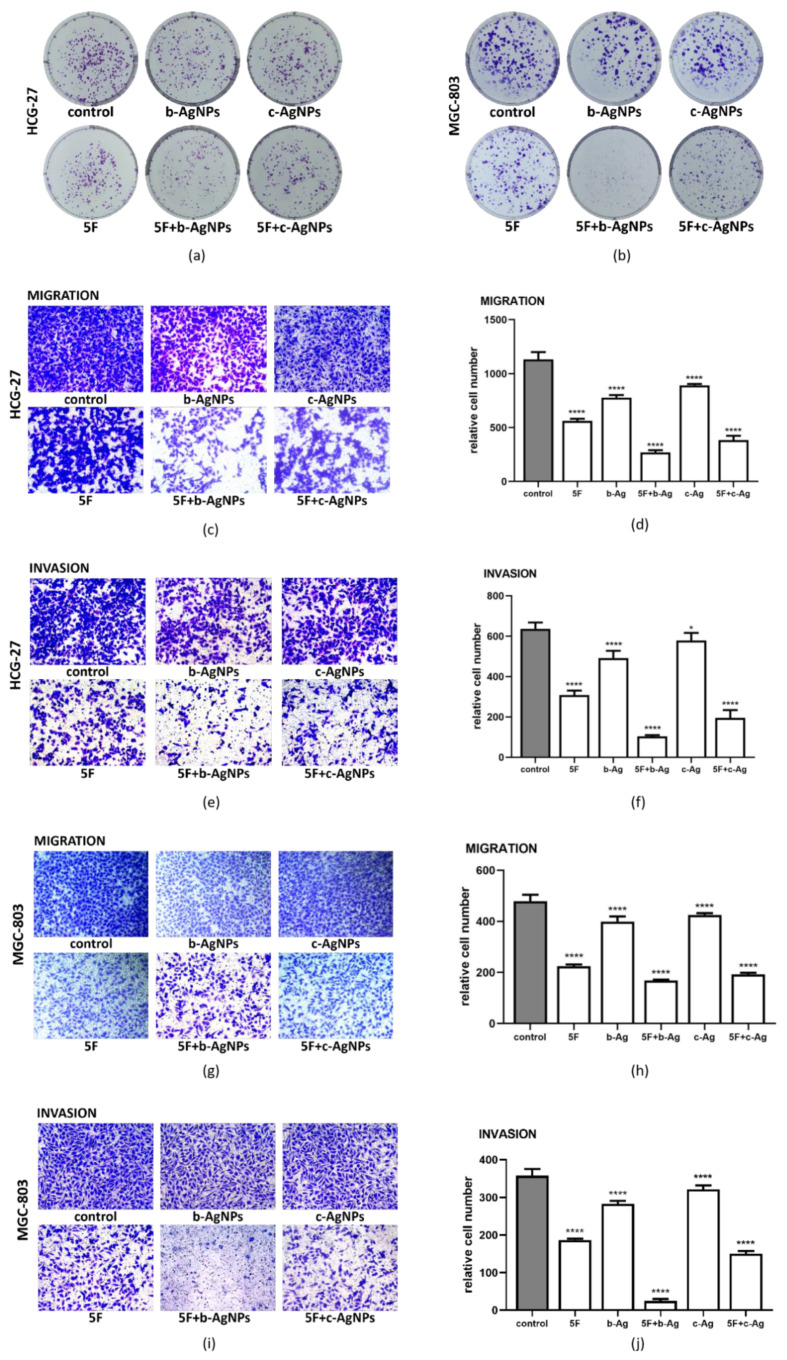
Proliferation, migration, and invasion of gastric cancer cells in different groups. (**a**) Results of colony formation assay in HCG-27; (**b**) results of colony formation assay in MGC-803; (**c**) results of migration experiment in HCG-27; (**d**) quantification of Figure 4c; (**e**) results of invasion experiment in HCG-27; (**f**) quantification of Figure 4e; (**g**) results of migration experiment in MGC-803; (**h**) quantification of Figure 4g; (**i**) results of invasion experiment in MGC-803; (**j**) quantification of Figure 4i. * *p* < 0.05, **** *p* < 0.0001.

**Figure 5 pharmaceutics-14-02109-f005:**
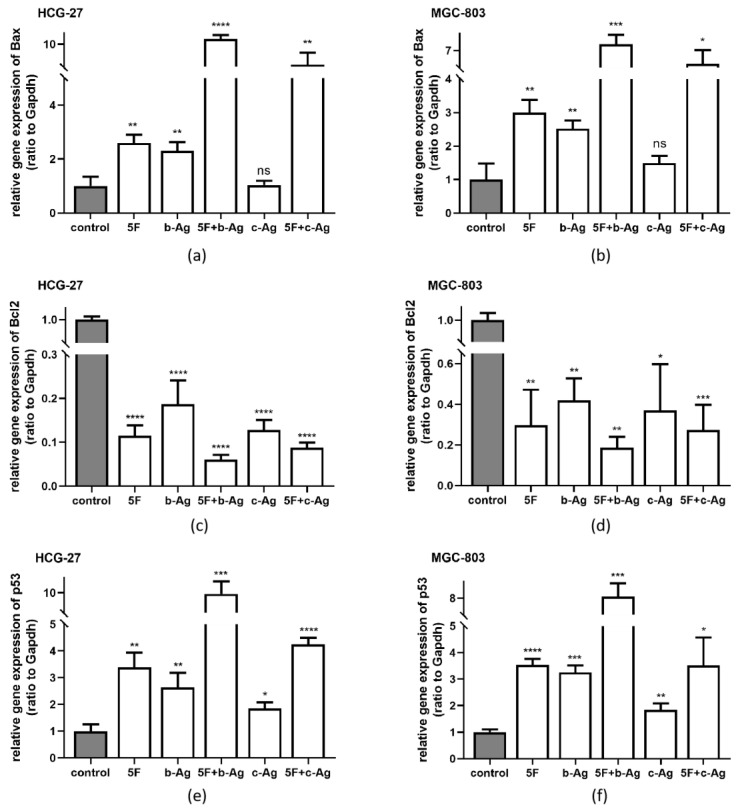
Bax, Bcl_2_, p53 mRNA expression level. (**a**) The mRNA expression of Bax in HCG-27; (**b**) the mRNA expression of Bax in MGC-803; (**c**) the mRNA expression of Bcl_2_ in HCG-27; (**d**) the mRNA expression of Bcl_2_ in MGC-803; (**e**) the mRNA expression of p53 in HCG-27; (**f**) the mRNA expression of p53 in MGC-803. * *p* < 0.05, ** *p* < 0.01, *** *p* < 0.001, **** *p* < 0.0001.

**Figure 6 pharmaceutics-14-02109-f006:**
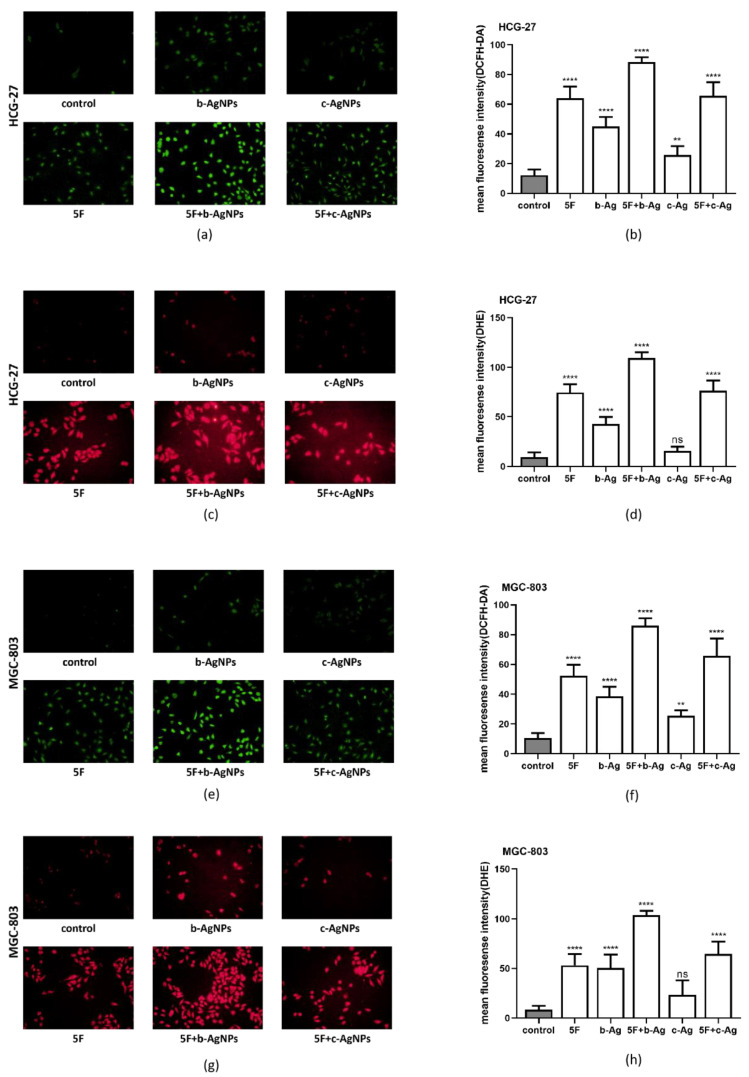
Mean fluorescence intensity of ROS probes in each group of cells. (**a**) Mean fluorescence intensity of DCFH-DA probe in HCG-27; (**b**) quantification of Figure 6a; (**c**) mean fluorescence intensity of DHE probe in HCG-27; (**d**) quantification of Figure 6c; (**e**) mean fluorescence intensity of DCFH-DA probe in MGC-803; (**f**) quantification of Figure 6e; (**g**) mean fluorescence intensity of DHE probe in MGC-803; (**h**) quantification of Figure 6g. ** *p* < 0.01, **** *p* < 0.0001.

**Table 1 pharmaceutics-14-02109-t001:** The concentration of b-Ag.

AgNO_3_ (μL)	AX (μL)	ddH_2_O (mL)	Concentration of b-Ag (PPm)
12.5	37.5	4.95	64.5
37.5	112.5	4.85	193.5
62.5	187.5	4.75	322.3
125	375	4.5	644.6
375	1125	3.5	1933.7

**Table 2 pharmaceutics-14-02109-t002:** The concentration of c-Ag.

CX (μL)	ddH_2_O (mL)	Concentration of c-Ag (PPm)
10	0.99	64.5
30	0.97	193.5
50	0.95	322.3
100	0.9	644.6
300	0.7	1933.7

**Table 3 pharmaceutics-14-02109-t003:** The concentration of 5F.

5F (μg)	PBS (mL)	Concentration of 5F (PPm)
0.34	1	64.5
1.02	1	193.5
1.7	1	322.3
3.4	1	644.6
10.2	1	1933.7
17	1	3222.9
34	1	6445.8

**Table 4 pharmaceutics-14-02109-t004:** Abbreviations of groups and concentrations.

Abbreviations	Drugs and Concentrations
Control	Without any drugs
5F	5-Fluorouracil (644.6 PPm)
b-Ag	Biosynthetic silver nanoparticles (322.3 PPm)
5F + b-Ag	5-Fluorouracil (644.6 PPm) + biosynthetic silver nanoparticles (322.3 PPm)
c-Ag	Chemically synthesized silver nanoparticles (322.3 PPm)
5F + c-Ag	5-Fluorouracil (644.6 PPm) + chemically synthesized silver nanoparticles (322.3 PPm)

## Data Availability

Not applicable.

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
