# Peer review of "Biosynthetic Silver Nanoparticles Inhibit the Malignant Behavior of Gastric Cancer Cells and Enhance the Therapeutic Effect of 5-Fluorouracil by Promoting Intracellular ROS Generation and Apoptosis"

_pharmaceutics, 2022, doi:10.3390/pharmaceutics14102109_

Round 1

Reviewer 1 Report

The reviewed manuscript has practical relevance, yielding interesting applications, worth publishing, once the authors address some aspects, which in my opinion require their attention.

Although the article is overall well-argued, there are some parts of the manuscript which require attention as to their syntax and grammar. I mention some indicative below:

·        There are several instances of run-on sentences, with random comma placement, odten confusing the primary points of the sentence.

·        Page 1, Line 35, “As emerging biotechnology, nanoparticles have been proved…” though the sentence starts with a subordinating conjunction, it does not seem to connect two clauses of unequal value to each other. It thus seem syntactically incorrect and needs rephrasing.

·        Similarly to the above, sentence lines 37-39 “Nanoparticles have also been paid more and more… cell-specific targeting” seems syntactically incorrect.

·        Page 1, Line 38, “…superior biocompatibility…” superior to what? I would argue that the biocompatibility of AgNPs in inferior to most (if not all) biomaterials. I would even say that it is erroneous to call AgNPs biomaterials!

There are more of these linguistic issues throughout the text and I’d ask the authors to carefully proofread their manuscript again and consult a colleague versed in scientific English writing.

Abbreviations should be explained on first appearance, e.g. CCK8 is firstly mentioned as a chapter title on page 2 line 90, prior to being explained on page 2 line 94..

Also, keywords are used to assist search engines to find the article, as such, is seems redundant to use keywords apparent in the title.

In the Abstract authors mention “Silver nanoparticles … due to their cytocompatibility…”. I don’t believe that cytocompatibility is a viable term for AgNPs, as these have been proven a cytotoxic by several studies. The authors themselves report reduced cell viability by 50% even at 5 μmol/L. Please remove the term.

Concerning the Materials and Methods section:

·        The authors refer to AgNPs concentrations in μmol/L. As the majority of studies in literature report ppm values, Id urge them to convert all figures to ppm instead, as to ease a comparison to other similar studies.

·        Cell groups mentioned in chapter 2.1 (e.g. 5F, b-Ag, 5F+ b-Ag…) should be explained immediately after reference, ideally with a table to provide better overview of the acronym’s meaning.

·        Page 2, line 70. The term “gift” is non-appropriate for a scientific article. Please use “provided by” instead.

·        Something seems off with the dimethyl sulfoxide/powder concentration (lines 76-77). How can 10mg of powder be suspended/diluted in 1mL of solution? Maybe 10μg instead? Please check carefully.

Results section

·        I think it is in-appropriate to use the conclusion of the results as a chapter title (e.g. lines 176-177 on page 4). The chapter title should reflect the content of the paragraphs that follow, not a summary thereof.

Please provide an approval protocol of the Ethics Committee.

Reviewer 2 Report

The authors (Yuan J. et al.) of the manuscript “Biosynthetic Silver Nanoparticles Inhibit the Malignant Behavior of Gastric Cancer Cells and Enhance the Therapeutic Effect of 5-Fluorouracil by Promoting Intracellular ROS Generation and Apoptosis” provide data on the ability of AgNPs to enhance the effect of 5-Fluorouracil (5F), an anticancer drug. The work was performed on human gastric cancer cells MGC-803 and HCG-27, primary obtained from human gastric carcinoma. The authors tested AgNPs fabricated using a chemical reducing agent (c-AgNPs) and a reducing agent obtained by the methanol extraction from Olax scandens leafs (b-AgNPs).

Major points:

The authors describe two methods to fabricated cAgNPs and b-AgNPs. Theses AgNPs demonstrated different cytotoxicity, assistant to antitumor effect of 5F, ability to induce apoptosis and ROS production etc. This is the focal points of the work. However, the paper does not present any characteristics of the obtained AgNPs.

Without these data, the main conclusion of the work "Biosynthetic silver nanoparticles inhibit and enhance 5F to inhibit the growth, migration, and invasion of gastric cancer cells by promoting apoptosis and increasing intracellular ROS production" is not valuable, and so could not accept.

Minor point:

Fig. 2. “c” and “d” are not present in the figure, it is not indicated, what 5F concentration is used for combinations with AgNPs.

Fig. 3 - 5. It is not indicated in what proportions 5F and AgNPs are used.

Fig. 3. The Y-axis is marked: relative cell number. How this value was calculated is not explained.

Reviewer 3 Report

Dear Editor of Pharmaceutics,

I reviewed the manuscript pharmaceutics-1850645 entitled “Biosynthetic Silver Nanoparticles Inhibit the Malignant Behavior of Gastric Cancer Cells and Enhance the Therapeutic Effect of 5-Fluorouracil by Promoting Intracellular ROS Generation and Apoptosis”.

The authors evaluated the capacity of biosynthetic silver nanoparticles (b-AgNPs) to cause cytotoxicity to gastric cancer cell lines and compared the effects to chemically synthesized AgNPs (c-AgNPs).

The findings demonstrated that biogenically synthesized AgNPs are more toxic than chemically AgNPs. Also, the combination of b-AgNPs with 5- Fluorouracil improves the cytotoxic effect of b-AgNPs. Thus, inhibiting cell viability, cell migration, and invasion ability and increasing ROS production.

I consider the manuscript needs to be grammatically reviewed; the methodology must also be written and reviewed carefully.

Importantly when working with nanomaterials is to show an extensive physicochemical characterization of the given nanomaterial. Some chemical and physical properties of the nanoparticles can strongly influence their biological effects. Therefore, it is necessary that the authors present a complete physicochemical characterization study of both b-AgNPs and c-AgNPs. And to discuss their role in cytotoxicity.

Considering all of the above, I would not recommend publishing the manuscript in its current form until the authors fulfill the following minor and major suggestions.

Major:

1.     It is necessary to present physicochemical characterizations of both b-AgNPs and c-AgNPs. In addition, it is crucial to report the size of both types of AgNPs and to present a TEM micrograph showing a representative population from the nanoparticles; UV-Vis, DLS: zeta potential, and hydrodynamic radio are also needed.

2.     It is also essential to compare the size and the size distribution of both nanoparticles in terms of their cytotoxicity. Perhaps the biogenic-AgNPs are more cytotoxic due to their small size. 

3.     Indicate how the concentration of AgNPs was calculated.

4.     Review whether the values for relative gene expression of Bax are correct. Ensure the values are correctly calculated due to the extremely high levels (almost 100 and 70 for HCG-27 and MGC-803 cells, respectively).

5.     What about silver ions released from nanoparticles? It is extremely important to add experiments to probe that the toxicity of b-AgNPs is higher than silver ions.

6.     What is the binding efficiency of the drug to silver nanoparticles? This is a critical study.

Minor:

1.     Consider referencing the paragraph in the abstract where it says: “Silver nanoparticles are increasingly used in diagnosis and treatment due to their cytocompatibility…”; Silver nanoparticles are well known to be cytotoxic; thus, add the proper reference to support this idea.

2.     The name of the species has to be in italics.

3.     The chemical formulae are incorrectly written, i.e., for CO2 and AgNO3

Reviewer 4 Report

We need to clarify some things before we can move forward

Ø There isn't much in-depth analysis of each section in this article.

Ø How do the author characterization of nanoparticles?. Could the author add additional statistically significant characterisations?

Ø What can we infer about placement from nanoparticles?

How to apply this material in vivo study?

Round 2

Reviewer 1 Report

Although the authors have addressed most of my comments, there are still some points which require their attention.

The evident lack of NPs characteristics. It is not enough to cite a previous article for those, especially since bio-synthesized silver nanoparticles tend to vary in-between different batches. Without data as to their shape, size distribution, surface load etc. it is impossible to evaluate the results with respect to prior literature.

Similar to above, there is another point which perplexes the comparison of the results presented here, to other similar studies. Referring to AgNPs concentrations in μmol/L instead of ppt hinders the direct comparison to 99% of existing literature using ppm as a reference value. I’d urge them to convert all figures to ppm instead, as to ease a.

Author Response

Response: Dear reviewer thanks to your valuable comments. We have taken your suggestion as you suggested. All the possible characterizations have been inserted in the manuscript via red highlights in different sections.

We have also converted the said unit i.e. μmol/L to ppm. See results section, Table 1 to Table 4.

Reviewer 2 Report

The authors gave satisfactory answers to my questions and comments, and also made the necessary corrections to the text of the manuscript.

Author Response

Response: Dear reviwer thank you for your valuable comments and affirmation.

Reviewer 4 Report

I want to give suggestion that to study in vivo animal study of the developed nanoparticles.

Author Response

Response: Dear reviewer thanks to your valuable comments. Basically in this time we have a limited facility and limited time because we have in hurry to accept and to secure my students degrees. In future we have a plan to use against vivo too.

Round 3

Reviewer 1 Report

The authors gave satisfactory answers to my questions and comments, and also made the necessary corrections to the text of the manuscript. I believe now it can be published in its present form.

Author Response

Dear reviwer thank you for your valuable comments and affirmation.
